# Facing the Forecaster's Dilemma: Reflexivity in Ocean System Forecasting

**Nicholas R. Record** [1,*] and **Andrew J. Pershing** [2]

1   Bigelow Laboratory for Ocean Sciences, East Boothbay, ME 04544, USA
2   Climate Central, Inc., Princeton, NJ 08542, USA; apershing@climatecentral.org
*   Correspondence: nrecord@bigelow.org

**Abstract:** Unlike atmospheric weather forecasting, ocean forecasting is often reflexive; for many applications, the forecast and its dissemination can change the outcome, and is in this way, a part of the system. Reflexivity has implications for several ocean forecasting applications, such as fisheries management, endangered species management, toxic and invasive species management, and community science. The field of ocean system forecasting is experiencing rapid growth, and there is an opportunity to add the reflexivity dynamic to the conventional approach taken from weather forecasting. Social science has grappled with reflexivity for decades and can offer a valuable perspective. Ocean forecasting is often iterative, thus it can also offer opportunities to advance the general understanding of reflexive prediction. In this paper, we present a basic theoretical skeleton for considering iterative reflexivity in an ocean forecasting context. It is possible to explore the reflexive dynamics because the prediction is iterative. The central problem amounts to a tension between providing a reliably accurate forecast and affecting a desired outcome via the forecast. These two objectives are not always compatible. We map a review of the literature onto relevant ecological scales that contextualize the role of reflexivity across a range of applications, from biogeochemical (e.g., hypoxia and harmful algal blooms) to endangered species management. Formulating reflexivity mathematically provides one explicit mechanism for integrating natural and social sciences. In the context of the Anthropocene ocean, reflexivity helps us understand whether forecasts are meant to mitigate and control environmental changes, or to adapt and respond within a changing system. By thinking about reflexivity as part of the foundation of ocean system forecasting, we hope to avoid some of the unintended consequences that can derail forecasting programs.

**Keywords:** ocean forecasting; reflexivity; endangered species; fisheries; harmful algal blooms; coupled natural-human systems; Anthropocene ocean

## 1. Introduction

The convention of studying natural systems—atmosphere, biosphere, etc.—as separate from human systems, is beginning to change [1]. Earth System Science and Anthropocene Studies increasingly couple natural and social science. As these disciplines merge, and especially in the context of applications, viewing scientists as part of a system of study is a challenge.

In forecasting applications within the natural sciences, this challenge has generally meant that a forecast is assumed to have no direct bearing on the outcome of the event in question. For example, forecasting rain does not affect whether or not it actually rains. Forecasting the time of a lunar eclipse does not alter the time of the eclipse.

In social sciences, on the other hand, it is more common for scientists to view predictions as a dynamic part of the subject system. For example, the prediction of a stock market collapse can itself cause a market collapse if investors panic in response to the prediction. This could happen even if no collapse would have occurred in the absence of the prediction [2]. This phenomenon is known as "reflexivity" and is a common component

of human systems (Figure 1). Reflexive prediction has been examined in many fields, such as economics, political science, and even studies of faith healing [3], and has a rich history of study in the social sciences [4,5]. Reflexivity itself is sometimes seen as the property that distinguishes the natural sciences from the social sciences [6,7].

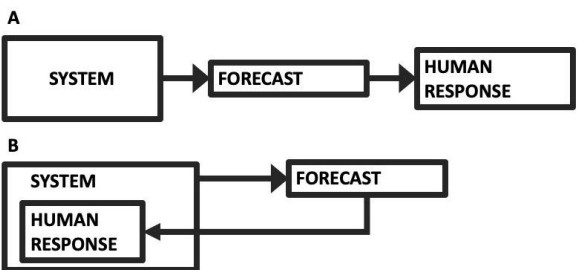

**Figure 1.** (**A**) The conventional forecasting scheme, where a system informs a forecast, which informs some human response. (**B**) A reflexive forecasting scheme where the human response is part of the system dynamics.

Natural systems forecasting has deep roots in weather forecasting, which is generally non-reflexive. However, many natural systems do have reflexive dynamics. For example, the dissemination of epidemic forecasts can alter human responses, changing the dynamics of the epidemic itself. A dire epidemic forecast could prompt a severe lockdown, thereby stifling the epidemic. Yet without the prediction, the lockdown might have come too late, and the dire outcome might have come to pass. There is evidence that the COVID-19 pandemic has reflexive dynamics and that taking these dynamics into account alters forecasts and outcomes [8].

Ocean system forecasting differs from weather forecasting in that many societally important forecasts deal with reflexive systems. Fisheries management often depends on a prediction of the stock size in future years. In turn, yearly fisheries forecasts can alter both fishing and management behavior, changing the mortality dynamics of the fish stocks. Similarly, endangered species management often relies on forecasts from population viability analysis. Management actions based on these forecasts are aimed at changing the predicted population trajectories. Even predictions of the global ocean climate system depend strongly on the human response to climate predictions themselves, where one of the explicit goals of making projections is to inform policy choices that will change the human forcing of the climate system.

Despite the prevalence of reflexivity in natural systems, it is typically not proactively taken into account in natural systems forecasting. The effects of reflexivity often lead to unexpected or unintended consequences, thus many fishery and endangered species management efforts require periodic yearly or multiannual iterative reevaluation of the objectives and targets, responding to missed targets reactively. For example, the Common Fisheries Policy of the European Commission has the requirement that any multiannual plan will "provide for its revision after an initial ex-post evaluation, in particular to take account of changes in scientific advice" [9]. Yet with increasingly more real-time forecasts available at people's fingertips, it is more urgent to understand reflexivity on a foundational level and to develop approaches to incorporate this understanding into forecasts proactively.

The feedback dynamic of reflexivity makes the phenomenon somewhat paradoxical, frustrating many attempts at prediction. A classic example is the 1948 U.S. election between Thomas Dewey and Harry Truman. Forecasts of a Dewey victory were so confident that the Chicago Tribune published the now infamous "Dewey Wins" headline on the day after the election. While it is possible that the mistaken forecast could have been due to a statistical or methodological error, the evidence suggests that the perception of a likely Dewey victory altered voter behavior [10]. That is, the forecast itself, and its dissemination, influenced (i.e., reversed) the outcome of the election. If that is true, then forecasts of a

Truman victory would have led to the opposite outcome—a Dewey win. The election was close—would it have been impossible to correctly forecast this event precisely because of the reversing effect of the forecast? This case illustrates the paradox of trying to make an accurate prediction in a reflexive system.

The paradoxical nature of reflexivity has been explored in literature, ranging from Oedipus to Ebenezer Scrooge to Asimov's Foundation, and has the same self-referential characteristic employed in philosophical cornerstones, such as Russell's Paradox and Gödel's Incompleteness Theorem [7]. The apparent paradox has led some to conclude that prediction in reflexive systems is not possible. The "Law of Forecast Feedback" [11] argues that a reliable prediction is not possible in a reflexive system. This pessimism is understandable, particularly when it comes to forecasting single binary or low-frequency events, such as elections or market collapses. Although natural sciences have often omitted reflexivity, they may offer an opportunity to address this paradox. Many natural systems forecasting programs involve high-frequency iterative forecasting, where forecasts are made and evaluated on a short time scale. The iterative nature of these applications provides an opportunity to examine how reflexivity works, and whether there are patterns that emerge or strategies that can be employed to make prediction successful despite reflexivity.

This paper examines the consequences of an iterative forecasting system having a reflexive component. It builds from a first-principles framework for prediction in ecology, adding a reflexive term to the dynamics. In particular, we incorporate two main elements of reflexive prediction: first, that the outcome would have been different without dissemination of the forecast, and second, that the forecast was believed and acted on [6]. We do not explicitly treat the mode of forecast dissemination. In practice, the mode of forecast dissemination is a key part of its influence on human behavior. For the purpose of illustrating some foundational properties of reflexivity in forecasting, we do not expand on the modes of forecast dissemination and the wide range of potential responses, but we recognize it as another important component to forecast implementation. By mapping previous ocean forecasting efforts into a biparametric time–time space, we explore how the iterative nature of many ocean forecasting endeavors can inform our understanding of reflexivity in forecasting, and we chart possible ways forward.

## 2. Theory

A generalized formulation of an ecosystem forecast can be written in terms of component parts as [12]:

$$Y_{t+1} = f(Y_t, X_t | \overline{\theta} + \alpha) + \epsilon_t \tag{1}$$

where $Y_t$ is the state variable we are trying to forecast for time $t + 1$, $X_t$ are environmental covariates, $\overline{\theta}$ and $\alpha$ represent model parameter mean and error, and $\epsilon_t$ represents process error. To analyze the effects of reflexive prediction in an iterative forecast, we will examine a simple example of this general formulation,

$$Y_{t+1} = \beta_1 Y_t + \beta_0 + \epsilon_t \tag{2}$$

This is the basic case for discontinuous (discrete) forecasting, where the state variable $Y$ at time $t + 1$ is a linear function of the previous time step—essentially a linear autocorrelative model. The simplified formulation allows us to explore basic properties of iterative reflexive prediction. The general idea can be extended to a more complex model. To account for reflexivity, we separate the actual system trajectory, $Y_t$, from the disseminated forecast, $Z_t$. As pointed out earlier, we don't explore different modes of dissemination here, but we note that different modes of dissemination can lead to different forms of reflexivity.

**Case 1: No reflexivity.** In the simple model system for $Y$, the best forecast equation would be

$$Z_{t+1} = \beta_1 Y_t + \beta_0 \tag{3}$$

where the disseminated forecasted $Z_{t+1}$ is identical to $Y_{t+1}$ without the error term. A forecaster could use this equation to make reliable and unbiased predictions at each

time step, with uncertainty described by $\epsilon$ plus whatever uncertainty exists around the parameter measurements. This case represents the conventional, non-reflexive point of view. The forecast has high accuracy, basically limited only by the magnitude of the error terms (Figure 2A).

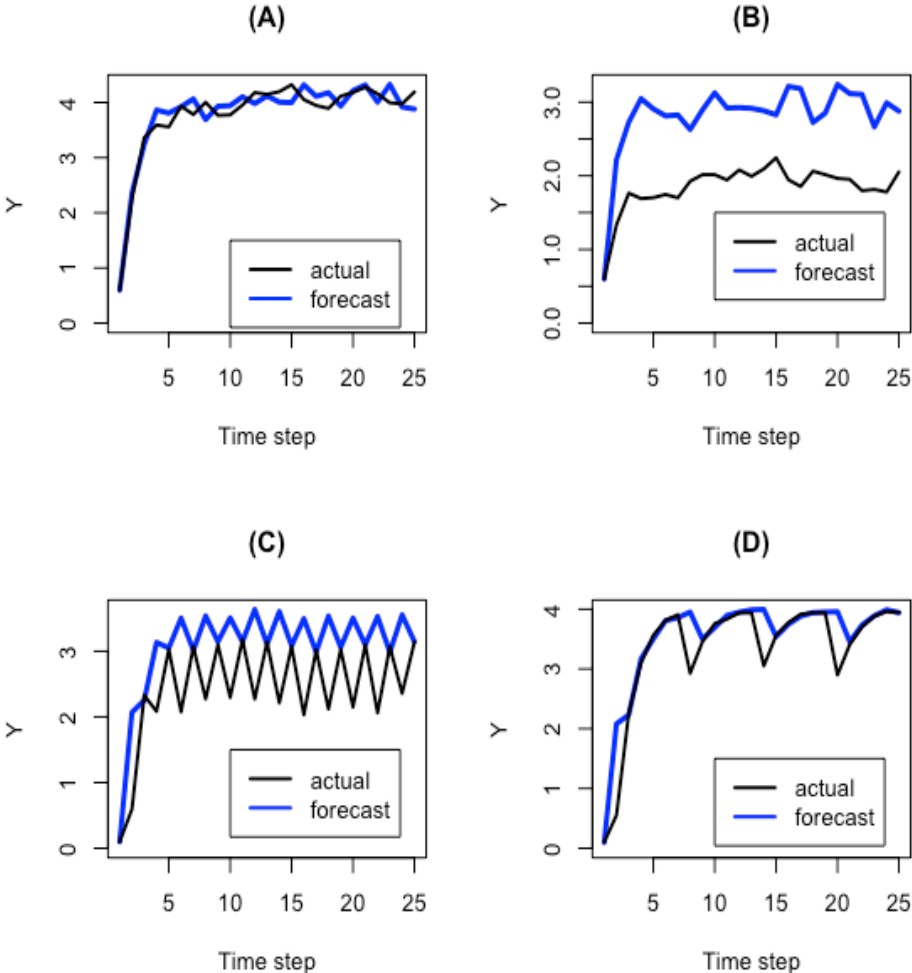

**Figure 2.** (**A**) Simulation with no reflexivity. (**B**) Simulation with reflexivity. (**C**) Simulation with reflexivity including a response to forecast accuracy. Error has been set to zero to make the cyclicity apparent. (**D**) Simulation with reflexivity including a response to forecast accuracy that includes memory of the accuracy over the past five time steps.

**Case 2: Self-defeating reflexivity**. In a reflexive prediction system, the outcome depends on the prediction. One way to express this is to add a reflexivity term to the general forecast equation:

$$Y_{t+1} = f(Y_t, X_t | \bar{\theta} + \alpha) + \epsilon_t + g(Z_{t+1}) \tag{4}$$

where $g$ is some function of the disseminated forecast. This function is analogous to the "internal decision model" [13]. Here, the outcome of the event at time $t + 1$ depends on what the forecast was for that time (i.e., $t + 1$). There are two types of reflexive prediction: self-fulfilling and self-defeating (also referred to as "bandwagon" and "underdog" [4]). In a self-fulfilling reflexive system, forecasting a particular outcome makes that outcome more likely (e.g., the market collapse example). In a self-defeating reflexive system, forecasting a particular outcome makes that outcome less likely (e.g., the Truman election). Here we take the self-defeating reflexive prediction as the illustrative case.

For self-defeating forecasts, $Y$ could be an index of some effect, such as the magnitude of an epidemic or the mortality rate of an endangered species—something that stakehold-

ers would generally want to minimize. Dissemination of the forecast causes an inverse response, decreasing the magnitude of $Y$. In the linear model example, we add a response term to the forecast equation:

$$Y_{t+1} = \beta_1 Y_t + \beta_0 + \epsilon_t - \rho_t(Z_{t+1}) \tag{5}$$

where $\rho_t$ is an increasing function of the disseminated forecast $Z_{t+1}$. The higher the forecast value $Z_{t+1}$ is, the stronger the counteracting response will be, leading to a smaller value of $Y_{t+1}$. The equation for $Z$ represents whatever the forecaster's strategy is for predicting the system. It could be the linear model that describes the non-reflexive $Y_t$, or something different.

While the scientific process behind a forecast aspires to objectivity, it exists in a broader system with subjective goals. These goals may be expressed in the kinds of processes that are forecasted (i.e., through funding choices) or in how forecasts are disseminated (e.g., the difference between forecasts of a hurricane path and communication strategies designed to get people out of harm's way). For the sake of simplicity, we will not separate the objective goals of scientific accuracy from the societal goals embedded in the application and dissemination of the forecast. Thus, when we refer to the forecaster's goals, we are essentially talking about the goals of the entire forecasting program.

Suppose the forecasting program were put in place with the ultimate aim of minimizing the value of $Y$ (i.e., stop the epidemic or eliminate the mortality rate of the endangered species). Under this reflexive scenario, the naive strategy would be to always provide the direct forecast. For this example, we formulate a response term: $\rho_t = \rho_0 \tanh(Z_{t+1})$. In other words, a high forecast value for the next time step motivates a response that counters the expectation. Using the tanh functional form caps the magnitude of the response. The forecast would have low accuracy, but the desired outcome would be achieved. On the other hand, a high-accuracy forecast would not prompt the response that minimizes the negative effect $Y$.

The consequences are twofold. First, by responding to the forecast as a warning, the actual value of $Y$ is driven down. This could be considered a case where the desired effect is achieved (Figure 2B). However, a second consequence is that the forecast is now never accurate. It always overshoots the actual by an interval equal to $\rho$ (on average). For this scenario to actually work, the forecast users would have to never catch on to the fact that the forecast is always more dire than reality. To put this into real terms, it would be akin to forecasting a fishery collapse every year, and although none ever occurs, the fishery repeatedly reduces catches as though a collapse were always imminent. This contradiction between forecast accuracy and forecast utility (from the perspective of the desired societal outcome) is the central point to the Law of Forecast Feedback.

**Case 3: Iterative self-defeating reflexivity**. Realistically, people would lose trust in a consistently dire forecast due to its consistent lack of accuracy. This is where the iterative dynamic comes into play. To account for this, we introduce a scaling factor $\tau$ to the response term that represents the reliability of the forecast:

$$Y_{t+1} = \beta_1 Y_t + \beta_0 + \epsilon_t - \rho_t \tau_t \tag{6}$$

Here $\tau$ is an inverse function of the error in the forecast (i.e., $\frac{|Z_t - Y_t|}{Y_t}$), where $\tau = 1$ when accuracy is perfect and goes to zero for very low-accuracy forecasts. In other words, as the forecast becomes inaccurate, it also loses its influence. Now we have the simplest fully iteratively reflexive forecast model.

We can formulate this factor $\tau$ as $\tau_t = e^{\frac{-\tau_0 |Z_t - Y_t|}{Y_t}}$. For a perfect forecast, $\tau_t = 1$, yielding a full response to the forecast. For very high inaccuracy, $\tau_t$ decays to zero, zeroing out the response term. The parameter $\tau_0$ shapes how quickly (as a function of forecast inaccuracy) the response term goes to zero. A high $\tau_0$ would mean that only a small amount of inaccuracy is needed for people to stop believing in and responding to the forecast. The

result is an oscillating pattern, where a reliable forecast is acted on, driving $Y$ down, thus making the next forecast inaccurate, diminishing the response, and driving $Y$ back up (Figure 2C). This is akin to the boom–bust reflexive dynamics seen in market systems [7].

**Case 4: Iterative + learning self-defeating reflexivity**. As a final note, there's no reason to assume that the response only depends on the previous time step. Depending on circumstances, it is possible that collective memory would evaluate the forecast reliability over multiple previous time steps. This can be added to the model using a number of time steps $m$, over which $\tau$ is computed and averaged. The result is a variably reliable forecast, with periodic lapses in accuracy (Figure 2D). From here, it is not difficult to imagine a wide range of periodic and quasi-periodic patterns that can occur depending on the form of $\tau_t$ and other properties of these equations. All of the richness of dynamical systems modeling could appear in the formulation of reflexivity.

### 3. The Forecaster's Dilemma

The question for the forecaster now becomes: how to deal with these opposing forces? On the one hand, a theoretically reliable forecast can alter behavior, making the forecast unreliable. On the other hand, consistently unreliable forecasts are likely to be ignored. The problem for the forecaster can be framed as the tension between two goals:

**Goal 1: The accuracy directive.** Conventionally, forecasters have tried to make predictions that accurately describe a future event. This also corresponds with goals of science to improve our understanding of the natural world. When the event comes to pass, a comparison between the forecast and the event serves as the assessment. This amounts to minimizing $\sum_t \frac{|Z_t - Y_t|}{Y_t}$.

**Goal 2: The influence directive.** The purpose of a forecast is usually to elicit some action. This generally corresponds with some practical societal goal. The $Y$ variable represents a negative effect that the forecast is aspiring to diminish over time, so this amounts to minimizing $\sum_t Y_t$ (This could also be framed as maximizing a positive effect, such as species recovery).

A forecaster in a reflexive system should consider whether it is possible to meet these two goals simultaneously, and if so, what is the best forecasting strategy i.e., the choice of function for $Z$ that accomplishes both directives? The example provided here is convergent in a recursive sense. That is, one can iteratively plug $Y_{t+1}$ back into the equation as $Z_{t+1}$, and the forecast for the next time step will converge on a value that is both accurate and minimizes the negative effect, basically toeing a line between the two cases. However, most real-world examples will probably be more complicated, with more dynamic and complex $g(Z)$ functions.

### 4. Solving the Forecaster's Dilemma

Reflexivity is not just of academic interest. The coronavirus pandemic brought home the point that reflexivity in forecasts can have very real consequences. As people come to use and expect increasingly more real-time forecasting, the issue of reflexivity represents an emerging scientific challenge. With the field of ocean system forecasting growing rapidly and experiencing a "Cambrian Explosion" in applications [14], forecasters should consider the role of reflexivity when adopting a forecasting program. We suggest three components to guide consideration of reflexivity. In some cases, dealing with reflexivity will be a nonissue. When reflexivity is deeply embedded within a system, however, new scientific and pragmatic approaches will need to be developed.

### 4.1. Step 1: Identify How Reflexivity Could Occur in the System

Ocean systems are organized across a range of temporal, spatial, and biological scales. These scales are related such that to first order, organisms that are small tend to have lower trophic level and shorter generation times and vice versa due to the strong size structuring in ocean ecosystems [15]. Ocean system forecasts tend to align along similar temporal, spatial, and biological scales because of this natural hierarchal structuring (Figure 3).

Most ocean system forecasting programs reviewed here fall along the one-to-one line in Figure 3. This scale matching is probably not an accidental coincidence, as decision making and predictability often align with the dominant process scales in a system. At one end of the spectrum, long-term planning for long-lived protected or exploited species is informed by their comparatively longer reproductive schedules, which aligns with a longer forecasting range [16,17]. Similarly, plankton with rapid doubling times fall at the other end of the spectrum, with shorter forecast ranges [18]. However, there are some examples that deviate from this one-to-one line. In the upper left region are harmful planktonic taxa like *Vibrio* and *Alexandrium*, which are short-generation taxa for which longer forecasts have been made [19,20]. This can be motivated either by forecasting and monitoring limitations, or by a particular forecast use. In the lower right, there are examples of short-range forecasts for protected or endangered taxa, such as whales and sea turtles. These few short-range forecasts for higher trophic levels focus on forecasting location rather than abundance, making the generation length less relevant. This also represents a response to a particular forecast use.

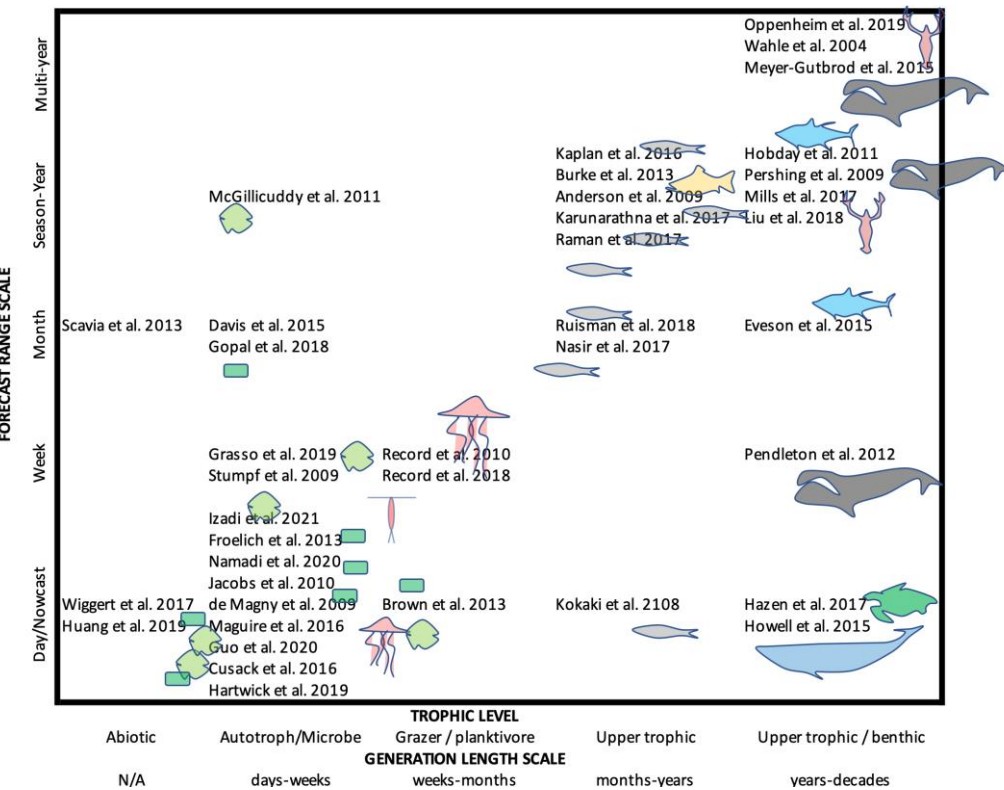

**Figure 3.** Forecast ranges of ocean system forecasting research programs and their associated biological time scales sources from literature examples, extended from the list provided in [14]. Positions are estimated based on descriptions in the texts [16–18,20–53].

This schematic needs to be shifted slightly to understand the role of reflexivity. Most of these forecasting studies do not consider the time scale of human response. Considering the coupled natural human system angle offers a slightly different lens, where we consider the dominant response time scales of the entire system, including the ecosystem and the human system together (Figure 4). If the response time is much longer than the forecast range (lower right of Figure 4), for example, reflexivity will be minimal to non-existent. In this scenario, many iterative forecasts would be made before any response takes place, so the human response would not affect the forecast. There are also cases where the human response has no bearing on the forecast. For example, a jellyfish forecast [46] might guide recreational activities, but probably would not influence the jellyfish populations themselves. Similarly, a forecast of the abundance of a harmful algal species might lead

to a fishery closure, but the bloom would persist unaffected. In any of these scenarios, if forecasting is guiding monitoring efforts, there is the possibility of feedback even in non-reflexive systems that can lead to confirmation bias.

If the coupled system response time is similar to or shorter than the forecast range, then there is the potential for reflexivity. These cases would fall near the one-to-one line or upper-left triangle of Figure 4. Examples include short-range forecasts of harmful or toxic algal species, or mid- to long-range forecasts for protected or exploited species. If there is the potential for reflexivity, the next step is to try to measure and describe it.

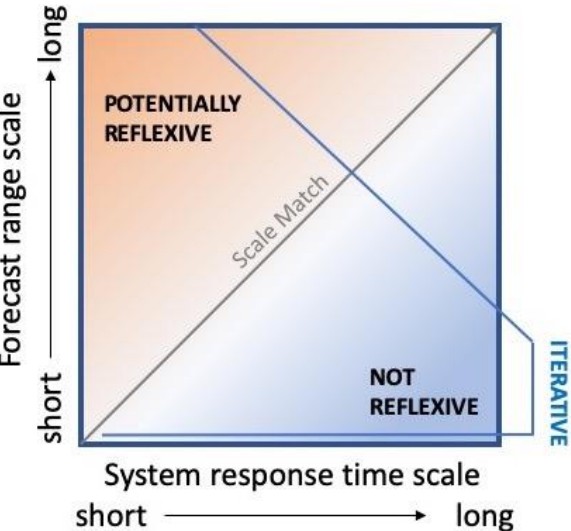

**Figure 4.** Schematic of forecast and system process scales. Here, the system includes the coupled natural human processes.

*4.2. Step 2: Determine Whether Reflexivity Is Self-Defeating, Self-Fulfilling, or a Combination of Both*

Once the potential for reflexivity has been characterized, the next step represents the empirical or data analysis side of the question. Detecting or measuring the human response can be challenging, especially if monitoring and modeling efforts have focused on the non-human parts of the system.

Consider, as an example, the management of endangered marine mammals. The Marine Mammal Protection act in the U.S. was amended in 1994 such that marine mammals are managed according to potential biological removal (PBR), which frames population viability as a forecast: i.e., "a certain probability of an event occurring in a given amount of time" [54,55]. The calculation of PBR includes a scaling factor $F_R$, which is set between 0 and 1. If a forecast looks particularly dire, setting $F_R$ to zero sets human-caused mortality to zero (in theory) regardless of the value of other factors. $F_R$ offers flexibility—and reflexivity—to the human response via the PBR calculation [56], and these numbers inform management decisions designed to reduce (or not) human-caused mortality of marine mammals. This places these forecasts toward the upper left of Figure 4, with long-range forecasts but short system response times.

If there is a way to quantify human response to the forecast, such as PBR, one can then examine whether the dynamics of this response is coupled to the dynamics of the natural system, such as in a dynamical systems framework. Two-way causality can be explored using tools such as Granger causality or convergent cross mapping [57], though there are cases where these tests are flawed [58]. In principle, it may be possible to detect reflexivity in ecological time series without measurement of the human response using Takens' theorem [59].

The species with the longest running PBR time series is the North Atlantic right whale, for which there are 25 years of data [60]. For the first decade of the 2000s, PBR was reduced

to zero by setting $F_R$ to zero in an attempt to restore this highly endangered species. The population climbed until 2010, at which point PBR was raised. Shortly thereafter, mortality rates began to rise (Figure 5A), and the species plummeted into crisis mode again [61]. When comparing PBR to human-caused mortality for this species, the strongest lagged correlation is with PBR leading mortality by four years (Figure 5B). Granger analysis supports this direction of lagged causality ($p < 0.05$, but see [58]), consistent with the notion that changing the forecast for the species has the following effect of changing the actual population trajectory of that species.

In some sense, this direction of causality shouldn't be surprising: the goal of the management approach is to have a following effect on the population. But the right whale example is a case of self-defeating reflexivity, and self-defeating reflexivity can cut two ways. A dire forecast can motivate recovery efforts, thereby improving the forecasted outlook, as intended—but a more favorable forecast can have the opposite effect. In the right whale time series, increases in PBR were followed a few years later by increases in human-caused mortality, reversing the recovery trend that the right whale population had been showing. While there are likely multiple factors at play in the sudden reversal of the right whale population trajectory, including climate and oceanographic changes [61,62], the pattern is consistent with the dynamics we would expect in a self-defeating reflexive forecasting system, leading to an unintended consequence. Ideally, we would like reflexive feedback to take effect when the population is struggling, but not when it's doing well—in other words, a concave-down curve in Figure 5B rather than concave up. Accounting for this kind of reflexive dynamic more deliberately probably requires a more mechanistic understanding of the reflexive term $g(Z)$.

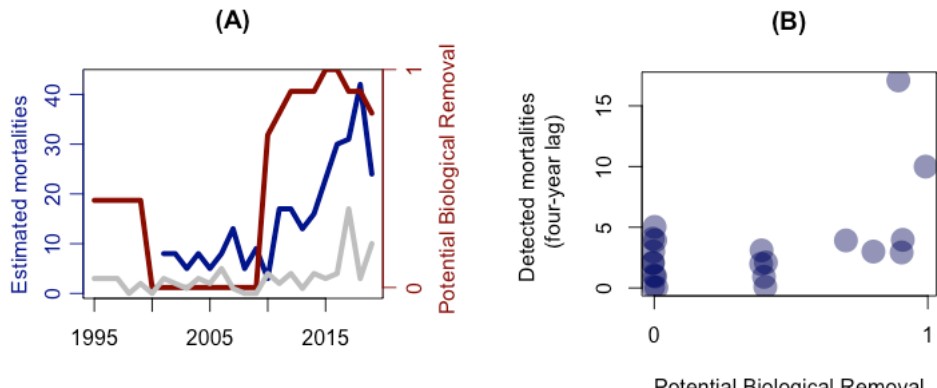

**Figure 5.** (**A**) Time series of PBR for the North Atlantic right whale (red) and two estimates of mortality: documented human-caused mortalities (grey) and the annual population change from the Pace model, subtracting out new calves (blue). (**B**) Lagged relationship between potential biological removal (PBR) and mortality. Data aggregated from NOAA and the North Atlantic Right Whale Consortium reports [60].

*4.3. Step 3: Incorporate Human Response into a Forecast Model*

If there is significant reflexivity in a forecasting system, with important consequences, the next step is to try to formulate that response mathematically and incorporate it into a model. This step represents an open area of scientific research and theory. A key question to answer here is: Can the accuracy and influence directives both be met?

Ocean forecasting up to now has largely followed the tradition of weather forecasting, combining mechanistic or processed based formulations, such as the advection-diffusion-reaction equation, with statistical formulations. More recently, machine learning algorithms have been replacing the statistical components, and to some extent the mechanistic components as well, to derive predictive rules from data. This framework has so far been mostly an elaboration of the $f(Z)$ term. The reflexive term $g(Z)$ represents a largely unexplored

research opportunity where similar approaches could be used. In fisheries forecasting, stochastic models have been used to couple these components [63].

Coupled natural-human systems are complex. There probably is not an analog for the Navier-Stokes equations for the human part of the system. However, examining mathematical formulations in a theoretical context can help answer whether the accuracy and influence directives are at odds with each other. If they are, then it could be a sign that the forecast will do more harm than good—or at least not have the intended effect.

There is also the opportunity to think beyond dynamical systems and statistics. For example, because information is exchanged in at least two directions in reflexive systems, some have viewed forecasting in the context of game theory [64]. This view becomes particularly interesting when multiple agents, including forecasters and users, are exchanging wide ranging information about the system in question. On the ocean, forecast users can be capturing, processing, and exchanging real-time observations and knowledge, such as where a commercial fish species is found—knowledge that is not available to the forecaster but plays a role in human response [65]. Similarly, multiple forecasters might be using different knowledge and approaches, and exchanging some part of that information with each other. In this way, forecasting programs can be nested within networks of social-natural systems with complex information flow.

The challenge of reflexive systems forecasting highlights the need to be making more operational forecasts. The iterative property of these forecasting systems is key, providing the data and experience needed to build up an understanding of reflexive dynamics. The tendency is often to focus forecasting efforts on high-stakes problems, such as endangered species or health hazards, but lower stakes problems (e.g., nuisance species, ecotourism) could provide a safer arena for building up the datasets needed to analyze and understand reflexive dynamics in ocean systems in new ways.

## 5. Conclusions: Reflexivity in the Changing Ocean

Reflexivity highlights the significant human dimension and associated challenges in emerging forecasting programs. Traditionally, whether forecasting the weather or some ecosystem process, the natural system is viewed in an objective sense, separate from the human observer. In the context of ocean systems, reflexivity is an emerging challenge that has bearing both to how we understand and interact with the ocean, and how we understand and make use of algorithms.

Regarding human interactions with the ocean, the "Anthropocene Ocean" is described as a socio-material space [66] where physical and biological systems are interlinked with social and scientific systems. In this context, there are two common frameworks useful for understanding ocean system forecasting. One framework is that of planetary security—mitigating the risks of environmental damage due to human activities. In this context, forecasting would serve as an aid to monitoring and controlling environmental processes. Reflexivity is implicit in this construct, as the human response is the mechanism for influencing the environment. Here both the accuracy and influence directives are important for the forecast to be effective.

With the urgency around issues like climate change, there are practical limitations to the "measure and control" approach to dealing with the Anthropocene. An alternative emerging perspective is the idea of correlational and relational epistemologies, where management structures would sense and adapt to events in real-time [66,67], without a causal understanding or attempt to mitigate the dynamics. This perspective also relies on algorithmic and digital technologies, but in this context, forecasting serves as information connectivity and not as a means of system control. Reflexivity is not necessarily implicit in this perspective. When reflexivity is present, a forecast could potentially still be useful without both of the two directives met, if it is nested within a larger network of correlational human responses and feedbacks [67]. This view has drawbacks as well, and potential for unintended consequences. As each new forecasting system comes online, we should ask whether it is intended to understand (in a causal sense) and control aspects of the ocean

system, or whether it is intended as part of a network of adaptation tools. This distinction will help narrow the scope of reflexivity in the forecasted system.

The question of whether the goal is to predict and control the environment, or to adapt and respond to a changing environment, is at the core of many discussions about big data, algorithms, and artificial intelligence. As forecasting algorithms become more widespread and embedded in our social relationship with Earth systems, ocean science can take lessons from the growing field of algorithmic accountability. Across applications ranging from resume sorting to prison sentencing, algorithms are replacing human decision making. The proliferation of algorithms in this way has led to many unintended consequences [68]. Ocean system forecasting shares this risk of unintended consequences—something that has already occurred in a few ocean forecasting programs [69,70]. For reflexive forecasts, when the accuracy and influence directives are at odds with each other, there is high potential for unintended consequences. The field of algorithmic accountability is developing methodologies for addressing this, such as action plans for redress when unintended outcomes occur, which can be applied to ocean forecasting to help prevent unintended consequences or address them when they occur [71–73].

Despite the potentially confounding nature of reflexivity, the subject represents a rich area of scientific inquiry. The reflexive term in the forecasting equation—i.e., the $g(Z)$—captures an emerging challenge in natural systems forecasting. Many forecasting evaluation analyses choose not to treat the reflexive feedback dynamic [74], and others have simply ignored it [75]. Some leave the human response to the realm of policy, communications, or to forecast users, while others view this part of the equation as a focus for quantitative study and analysis in its own right [2]. The example developed here separates $f(Y)$ and $g(Z)$ into additive terms, but it is possible that they interact in more complex and nonlinear ways yet to be discovered. There is also another layer of complexity added to $g(Z)$ dynamics when considering the mode of forecast dissemination. People respond differently depending on how a forecast is communicated, and if the system is reflexive, then communication choices can feed back on the natural system dynamics $f(Z)$. By representing iterative system forecasting as a combination of two components, $f(Y)$ and $g(Z)$, we see a promising quantitative starting point for integrating natural sciences with social and behavioral sciences, as well as a pathway for using forecasts as a tool for navigating the complex interactions between humans and the ocean.

**Author Contributions:** Conceptualization, N.R.R. and A.J.P.; methodology, N.R.R.; software, N.R.R.; validation, N.R.R.; formal analysis, N.R.R.; investigation, N.R.R. and A.J.P.; resources, N.R.R. and A.J.P.; data curation, N.R.R.; writing—original draft preparation, N.R.R. and A.J.P.; writing—review and editing, N.R.R. and A.J.P.; visualization, N.R.R. and A.J.P.; supervision, N.R.R. and A.J.P.; project administration, N.R.R. and A.J.P.; funding acquisition, N.R.R. and A.J.P. All authors have read and agreed to the published version of the manuscript.

**Funding:** Support for this research came from institutional funds from the Tandy Center for Ocean Forecasting at Bigelow Laboratory for Ocean Sciences and the Otto Mønsteds Fond via Denmark Technical University (NRR) and from NSF grant OCE-1851866 (AJP).

**Institutional Review Board Statement:** Not applicable.

**Informed Consent Statement:** Not applicable.

**Data Availability Statement:** Code and data used in this study are stored in the following open repository: https://github.com/SeascapeScience/forecasters-dilemma (accessed on 1 November 2021).

**Conflicts of Interest:** The authors declare no conflict of interest.

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
