# Peer review of "Facing the Forecaster’s Dilemma: Reflexivity in Ocean System Forecasting"

_2673-1924, doi:10.3390/oceans2040042_

Round 1

Reviewer 1 Report

Review of

Solving the forecaster’s dilemma: reflexivity in ocean system forecasting

By Nicholas R. Record, Andrew J. Pershing

Main point of the paper

The authors describe, that reflexivity has implications for many ocean forecasting applications, such as fisheries management, endangered species management, toxic and invasive species management, as the forecast could be part of the system (but only when actions are taken!). For dealing with the problem they propose a basic theoretical skeleton in order to consider reflexivity in ocean forecasting in an iterative way.

General comments:

In my understanding the general conclusion that an accurate prediction in reflexive systems is not possible is certainly the correct one. This is based in the very nature of such forecasts, as the agents in the system having “free will” could act on the forecast or not, again depending on many additional external factors. I agree with the authors that it might be however possible to arrive at approximate estimates using iterative (short term) forecasting of processes with reflexivity, thereby solving the dilemma.

However the authors are ignoring actual practice at least as it is commonplace in Europe. For example the Common Fisheries Policy of the European Commission requires the annually repeated adaptation of stock specific catch quota based on scientific advice (stock modeling and fishing mortality below MSY). This process seems to implement the author’s proposal rather exact. Similar, scientific – policy iterative interaction processes are ongoing in many more environmental legislations in Europe, as for example in implementing the Marine Strategy Framework Directive, based on a 6 years cycle of impact assessment and review. Ocean modeling is supporting this cycle especially the assessment of proposed measures (their effectiveness), thereby not only altering the forecast but potentially also requesting different measures, again altering the forecast.

So for me it seems obvious that the authors correctly identify the problem and a solution, but completely miss that this is already common procedure in Europe!

Despite the for me already achieved practical implementation of the concept in European policy (supported by science and ocean modeling), there is however seemingly no real scientific literature published, that is documenting the problem and proposing potential solutions (at least my non exhaustive search for reflexivity or forecast feedback did not bring up something useful).

Therefore a substantial improved paper could potentially fill that gap and could become a kind of scientific reference for practitioners.

Critical issues, misunderstandings:

Line17 (only as example)

understand whether forecasts are meant to mitigate and control environmental changes, or to adapt and respond within a changing system

This is at least imprecise language (potentially even a misunderstanding of the authors), as in my understanding “forecasts” are not meant to “mitigate or control” changes, they can be only of help to design and implement (or not), measures for mitigation or adaptation.

Unfortunately this wrong jargon is used more often in the paper, ignoring that after the forecast the critical question is how to act/react on the forecast. Do we ignore the forecast; do we implement measures, which measures?

As the appropriate action (measures taken) on the forecast is extreme relevant, its neglection in the paper, for me questions the validity of the full paper. Why have the authors ignored this aspect?

Line 106 (and 116)

I see a problem with the time indices in equation like: Y(t+1) ~ g Z(t+1), because when written like this, the forecast Z would act immediately on the real output Y in the same time step. This is however very unlikely to happen, because in the reflexivity process there are agents involved who need time to act. The forecast can only have impact on the next output time step.

Line 240

Two-way causality can be explored using tools like Granger causality or convergent cross mapping

It was proven by Liang that Granger causality has a mathematical flaw of not assuring nil for cases of nil causality, so the use of Granger causality must be avoided. Please refer to modern and mathematically correct causality concepts.

(see for example causality derived on the information flow concept)

https://journals.aps.org/pre/abstract/10.1103/PhysRevE.94.052201

271-272

More recently, machine learning algorithms have been replacing the statistical components, and to some extent the mechanistic components as well. This framework is essentially an elaboration of the ?(?) term.

This statement points to a potential misunderstanding of the term “machine learning” by the authors. Machine learning is defined as learning the “rules from the data”, what implies that for sure both f(Z) and g(Z) would be learned from the data, if present. Obviously the precondition is, that the reflexivity must be present in the data, with sufficient abundance and accuracy.

Summary

Despite the paper is based on a correct general idea and is proposing a potential correct solution to the problem, I tend for rejection of the paper. However I could be willing to reconsider a new version based on a major revision.

Based on my understanding, the reflexivity problem in forecasting has been identified long ago and the proposed solution has been implemented since many years in actual European policy and legislation. Further the authors seemingly missed the main point about forecasting, namely the action on the forecast, ignoring the question if mitigation or adaptation measures will be implemented or not. In summary, despite some good intentions/ideas, for me the authors are about 20 years late with their proposal.

On the other hand it seems, however, no similar paper describing the theoretical problem and a potential solution is present in the literature.

Therefore a substantial improved and rewritten paper, covering also actual practical approaches to deal with the reflexivity problem, considering also the implementation of measures, could potentially fill that gap and could become a kind of scientific reference for practitioners.

Author Response

The reviewer comments are in italics below, and our responses are in normal text

General comments:

In my understanding the general conclusion that an accurate prediction in reflexive systems is not possible is certainly the correct one. This is based in the very nature of such forecasts, as the agents in the system having “free will” could act on the forecast or not, again depending on many additional external factors. I agree with the authors that it might be however possible to arrive at approximate estimates using iterative (short term) forecasting of processes with reflexivity, thereby solving the dilemma.

However the authors are ignoring actual practice at least as it is commonplace in Europe. For example the Common Fisheries Policy of the European Commission requires the annually repeated adaptation of stock specific catch quota based on scientific advice (stock modeling and fishing mortality below MSY). This process seems to implement the author’s proposal rather exact. Similar, scientific – policy iterative interaction processes are ongoing in many more environmental legislations in Europe, as for example in implementing the Marine Strategy Framework Directive, based on a 6 years cycle of impact assessment and review. Ocean modeling is supporting this cycle especially the assessment of proposed measures (their effectiveness), thereby not only altering the forecast but potentially also requesting different measures, again altering the forecast.

So for me it seems obvious that the authors correctly identify the problem and a solution, but completely miss that this is already common procedure in Europe!

            The Common Fisheries Policy, as well as other fisheries management approaches, is a productive approach, and we’ve added a discussion of in the text. The CFP approach is iterative in that it calls for revision after an initial ex-post evaluation, in particular to take account of changes in scientific advice. This allows for corrections, but it doesn’t provide a way to account for reflexivity proactively. For example, in all fisheries management systems that we are aware of, there is no explicit consideration of how fishing behavior will change in response to a particular choice of catch limits nor any evaluation of the likelihood that the actual catch will be above or below the target. The iterative approach we describe (as an illustrative example) is somewhat different from the CFP strategy in that the iterations are calculated until convergence is reached at each time step before moving on to the next time step. We’ve tried to make this distinction clearer in the text. We also don’t mean to imply that this is the only approach—convergence isn’t always possible, and often times there is too much uncertainty in the human response for this to be practicable. To this end, we discuss many approaches to dealing with reflexivity. But if we had a better understanding of the reflexive dynamics of a given system, there is the possibility of taking this into account proactively, rather than having to make corrections reactively.

Despite the for me already achieved practical implementation of the concept in European policy (supported by science and ocean modeling), there is however seemingly no real scientific literature published, that is documenting the problem and proposing potential solutions (at least my non exhaustive search for reflexivity or forecast feedback did not bring up something useful).

Therefore a substantial improved paper could potentially fill that gap and could become a kind of scientific reference for practitioners.

            We’ve made the improvements suggested by the reviewers, and hopefully this paper can be one step forward in filling the literature gap referenced here.

Line17 (only as example)

“understand whether forecasts are meant to mitigate and control environmental changes, or to adapt and respond within a changing system”

This is at least imprecise language (potentially even a misunderstanding of the authors), as in my understanding “forecasts” are not meant to “mitigate or control” changes, they can be only of help to design and implement (or not), measures for mitigation or adaptation.

Unfortunately this wrong jargon is used more often in the paper, ignoring that after the forecast the critical question is how to act/react on the forecast. Do we ignore the forecast; do we implement measures, which measures?

As the appropriate action (measures taken) on the forecast is extreme relevant, its neglection in the paper, for me questions the validity of the full paper. Why have the authors ignored this aspect?

            The distinction between the forecast and the action that is based on the forecast is an important one, as the reviewer notes. We tried to spell that out in the initial paper submission, but we’ve made the distinction clearer in this version of the text. The way forecast users respond to forecasts is indeed an interesting and open area of research.

Line 106 (and 116)

I see a problem with the time indices in equation like: Y(t+1) ~ g Z(t+1), because when written like this, the forecast Z would act immediately on the real output Y in the same time step. This is however very unlikely to happen, because in the reflexivity process there are agents involved who need time to act. The forecast can only have impact on the next output time step.

            In this formulation, the forecast (potentially) has an impact on the environment in the next output time step. In other words, the forecast is made for time step t+1 so that human agents have a chance to take action in advance of that time step. Thus the Z(t+1) forecast has the potential to affect the Y(t+1) environment. Of course, there are many different time frames at which these two processes may or may not be interacting (which we elaborate on in the paper), but this is the simplest scenario, which is the purpose of the model illustration. We’ve added some text to make this clearer.

Line 240

“Two-way causality can be explored using tools like Granger causality or convergent cross mapping”

It was proven by Liang that Granger causality has a mathematical flaw of not assuring nil for cases of nil causality, so the use of Granger causality must be avoided. Please refer to modern and mathematically correct causality concepts.

(see for example causality derived on the information flow concept)

https://journals.aps.org/pre/abstract/10.1103/PhysRevE.94.052201

            We have added this reference and a note of caution. We’ve been careful in our language not to conclude causality, but to look for lines of evidence.

271-272

“More recently, machine learning algorithms have been replacing the statistical components, and to some extent the mechanistic components as well. This framework is essentially an elaboration of the ?(?) term.”

This statement points to a potential misunderstanding of the term “machine learning” by the authors. Machine learning is defined as learning the “rules from the data”, what implies that for sure both f(Z) and g(Z) would be learned from the data, if present. Obviously the precondition is, that the reflexivity must be present in the data, with sufficient abundance and accuracy.

            The statement cited here is simply a comment on how machine learning is currently used in this context. We have added text in line with the reviewer’s comment suggesting that machine learning could be used in the g term as well (though this might not always be advised).

Summary

Despite the paper is based on a correct general idea and is proposing a potential correct solution to the problem, I tend for rejection of the paper. However I could be willing to reconsider a new version based on a major revision.

Based on my understanding, the reflexivity problem in forecasting has been identified long ago and the proposed solution has been implemented since many years in actual European policy and legislation. Further the authors seemingly missed the main point about forecasting, namely the action on the forecast, ignoring the question if mitigation or adaptation measures will be implemented or not. In summary, despite some good intentions/ideas, for me the authors are about 20 years late with their proposal.

            As noted above, the fisheries example is relevant, but not quite what we’re getting at with this review/perspective piece. We also don’t mean to argue that there is a single approach that will work across all of the marine systems that we reviewed. We added a brief discussion of the CFP, but our hope is to draw attention to the research gap (as the reviewer pointed out) and to stimulate further research on this topic, rather than to advocate for a particular solution.

On the other hand it seems, however, no similar paper describing the theoretical problem and a potential solution is present in the literature.

Therefore a substantial improved and rewritten paper, covering also actual practical approaches to deal with the reflexivity problem, considering also the implementation of measures, could potentially fill that gap and could become a kind of scientific reference for practitioners.

In this paper, we try to make the case for the need for foundational research on reflexivity in ocean system forecasting—i.e. the development of theory, monitoring, and modeling that improves our understanding of how reflexivity works. Implementation would be a subsequent step. Along these lines, one of the other reviewers suggested removing the word “solving” from the title so as not to imply that the paper offers a specific solution. We have taken this suggestion, and hopefully this makes the intention of the paper more transparent. In addition, we’ve rewritten and improved portions of the paper to better capture this intention and address the reviewer concerns.

Reviewer 2 Report

Line 83、87、95、106、116、145,order number (1)、(2)...should be added at end of each equation. 

Line 205, Figure 3 needs to be improved. 

Author Response

The reviewer comments are in italics below, and our responses are in normal text  

Comments and Suggestions for Authors

Line 838795106116145order number (1)(2)...should be added at end of each equation. 

            We have made this correction.

Line 205, Figure 3 needs to be improved. 

            We have made improvements to figure 3, including some new references and minor modifications.

Reviewer 3 Report

The manuscript Solving the forecaster’s dilemma: reflexivity in ocean system forecasting by Record & Pershing provides a fundamental framework for addressing the reflexivity in ocean forecasting. With the growing number and scope of forecasting systems in ocean science (climate change, conservation, fishery management, etc.) the problem of the reflexivity is becoming more and more urgent to include in the forecasting system. The manuscript sets up the framework, but of course does not provide any operative solution, nor it was intended to do it. Nevertheless, for this resaon, I would suggest the authors to change the title from "Solving the forecaster's dilemma:" to "Addressing the forecaster's dilemma" or "Introducing the forecaster's dilemma". Further, not all types of relationships between the forecasting and the reflexivity are investigated, nor is the problem of the different types of forecast dissemination and their influence on the reflexivity dynamic (as correctly pointed out by the authors themselves in Conclusion). Still, the manuscript is important in that it opens up the discussion for the inclusion of reflexivity in forecasting systems and shows the way on how to do it, thus, in my opinion, deserves to be published.

Author Response

The reviewer comments are in italics below, and our responses are in normal text.

Comments and Suggestions for Authors

The manuscript Solving the forecaster’s dilemma: reflexivity in ocean system forecasting by Record & Pershing provides a fundamental framework for addressing the reflexivity in ocean forecasting. With the growing number and scope of forecasting systems in ocean science (climate change, conservation, fishery management, etc.) the problem of the reflexivity is becoming more and more urgent to include in the forecasting system. The manuscript sets up the framework, but of course does not provide any operative solution, nor it was intended to do it. Nevertheless, for this resaon, I would suggest the authors to change the title from "Solving the forecaster's dilemma:" to "Addressing the forecaster's dilemma" or "Introducing the forecaster's dilemma".

This is a correct read of the intention of the paper, and we’ve followed this suggestion, removing “solving” from the title.

Further, not all types of relationships between the forecasting and the reflexivity are investigated, nor is the problem of the different types of forecast dissemination and their influence on the reflexivity dynamic (as correctly pointed out by the authors themselves in Conclusion). Still, the manuscript is important in that it opens up the discussion for the inclusion of reflexivity in forecasting systems and shows the way on how to do it, thus, in my opinion, deserves to be published.

Round 2

Reviewer 1 Report

I found the new paper submission substantially improved, so that I now recommend publication.